# Metagenomic Analysis of Antarctic Biocrusts Unveils a Rich Range of Cold-Shock Proteins

**DOI:** 10.3390/microorganisms11081932

**Published:** 2023-07-28

**Authors:** Ekaterina Pushkareva, Josef Elster, Burkhard Becker

**Affiliations:** 1Department of Biology, Botanical Institute, University of Cologne, Zulpicher Str. 47B, 50674 Cologne, Germany; b.becker@uni-koeln.de; 2Institute of Botany, Academy of Sciences of the Czech Republic, Dukelska 135, 37982 Trebon, Czech Republic; josef.elster@ibot.cas.cz; 3Centre for Polar Ecology, University of South Bohemia, Na Zlate Stoce 3, 37005 Ceske Budejovice, Czech Republic

**Keywords:** metagenomic sequencing, biocrusts, Antarctica, cold-shock proteins, antifreeze proteins, cold-shock domain

## Abstract

Microorganisms inhabiting Antarctic biocrusts develop several strategies to survive extreme environmental conditions such as severe cold and drought. However, the knowledge about adaptations of biocrusts microorganisms are limited. Here, we applied metagenomic sequencing to study biocrusts from east Antarctica. Biocrusts were dominated by cyanobacteria, actinobacteria and proteobacteria. Furthermore, the results provided insights into the presence and abundance of cold shock proteins (Csp), cold shock domain A proteins (CsdA), and antifreeze proteins (AFP) in these extreme environments. The metagenomic analysis revealed a high number of CsdA across the samples. The majority of the Csp recorded in the studied biocrusts were Csp A, C, and E. In addition, CsdA, Csp, and AFP primarily originated from proteobacteria and actinobacteria.

## 1. Introduction

Biological soil crusts (biocrusts) are communities of organisms (microbial phototrophs, heterotrophic bacteria, archaea, fungi, lichens, and mosses) living in the uppermost layer of soil (around 5–10 mm). They contribute to the nutrient and carbon availability, soil formation, and establishment of other organisms. Biocrusts are particularly important in extreme environments where only a few species of vascular plants are present. In these habitats, they are often the only or main primary producers [1].

Microorganisms inhabiting Antarctic biocrusts are either cold tolerant or psychrophiles as they are capable to live below 0°C. They acquire special proteins that prevent cell damage during freezing [2]. For example, antifreeze proteins (AFP) lower the water’s freezing point and avoid the growth of ice crystal in the cell [3]. Similarly, cold shock proteins (Csp) enable efficient transcription and translation during the cold shock [4]. Furthermore, cold-tolerant microorganisms and psychrophiles produce extremozymes, which are able to catalyze chemical reactions under harsh environmental conditions.

The majority of studies investigating Csp, CsdA, and AFP in microorganisms primarily focus on bacteria [3,4], while knowledge about these proteins in algae is limited to a few studies [5,6]. Similarly, the role of these proteins in soil ecosystems, including biocrusts, has been largely overlooked.

Here, we characterized biocrust metagenomes from Enderby Land and Queen Maud Land, Antarctica. Additionally, we provide an overview of cold shock and antifreeze proteins in biocrusts from east Antarctica, which could serve as a baseline for future research on microorganism adaptation in extremely cold environments.

## 2. Materials and Methods

Biocrust samples from in Enderby Land and Queen Maud Land in east Antarctica were collected in austral summer of 2018/2019 during the Japanese Antarctic Research Expedition (JARE60). The coldest month in these regions is August with an average air temperature of around −19 °C and the warmest is January with an average air temperature of around −1 °C. Five biocrusts samples were collected from four different localities: Amundsen Bay (Amu8 and Amu14), Langhovde Hills (Lang37), Skarvsnes Foreland (Skar18), and Syowa Station (Syo6). The detailed description of the collected samples is presented in Appendix A.

The pH and conductivity of two technical replicates per each sample were evaluated in demineralized and distilled water, respectively. Total DNA was extracted from each sample using the DNeasy PowerSoil Pro Kit (QIAGEN, USA) according to the manufacturer’s instructions. The extracted DNAs were sent to Eurofins Genomic (Germany), where metagenomic sequencing on an Illumina MiSeq platform was performed. The raw reads were submitted to the Sequence Read Archive (SRA) under the project PRJNA945601.

Bioinformatic analysis was performed in the OmicsBox software (v. 3.0.30) using standard settings [7]. Preprocessing of FASTQ files was conducted in Trimmomatic [8] and the rRNAs were separated from the dataset using SortMeRNA [9]. The taxonomic assignments of the extracted rRNAs were performed using Kraken 2 (v2.1.2; [10]). The remaining reads were assembled de novo for each sample separately using MEGAHIT (v1.2.8; [11]). Gene prediction based on open reading frames (ORFs) was conducted using FragGeneScan [12]. Functional annotations of novel sequences were further performed using precomputed eggNOG-based orthology assignments [13]. The contigs were also aligned to the NCBI Blast searches (E^−10^) and Gene Ontology (GO) mapping and annotations were performed [14].

In addition, sequences annotated as cold-shock domain A, cold-shock proteins, and antifreeze proteins were extracted from the dataset and taxonomically assigned using Kraken 2.

## 3. Results and Discussion

Metagenomic sequencing produced around 744 M of quality filtered reads. Of these, 0.2–0.3% were rRNAs. Around 4 M ORFs per sample were predicted with an average length of 369–462 bp (Table 1). Taxonomic analysis of the metagenomic data using Kraken2 revealed the presence of bacteria (64–83% of reads), eukaryotes (0.5–6% of reads), archaea (up to 0.3% of reads), and viruses (up to 0.03% of reads), and 16–42% of reads were not classified (Appendix A). Furthermore, the dominant bacterial phylum in the Lang37 and Syo6 samples was cyanobacteria, which is consistent with the results of 16 S rRNA sequencing Contrary to the results of amplicon sequencing, the samples Amu8 and Amu14, analyzed by metagenomic sequencing, exhibited a dominance of actinobacteria, while Skar18 had a higher number of reads assigned to proteobacteria. Previous comparative analysis between these two sequencing techniques demonstrated that metagenomics is a more preferable tool for evaluating microbial community structure when compared to amplicon sequencing [15].

Only 4% of the assembled contigs was annotated to GO by EggNOG (Table 1). The sequences were annotated into four categories: Information Storage and Processing (21% in average), Cellular Processes and Signaling (22% in average), Metabolism (38% in average), and Poorly Characterized (19% in average). No major differences in the proportions of these categories were observed among the sites. In contrast, between 29 and 30% of contigs were annotated to GO with Blast2GO implemented in OmicsBox software. Even though EggNOG-mapper performs fast functional annotation, the database is limited, especially for psychrophilic organisms. For example, antifreeze proteins, which will be discussed later, were annotated by Blast2GO, but were not found in the EggNOG dataset. Therefore, we used Blast2GO annotations for further results and discussions.

Cold-shock proteins (Csp) are known to help organisms to survive a rapid temperature drop and maintain physiological performance during a cold stress episode. Here, we recorded between 194 to 439 genes assigned to different Csp subfamilies in the studied biocrusts (Figure 1, Appendix A), which represents a higher number of genes compared to previous studies on Antarctic soil [16]. CspA, C, and E were the dominant subfamilies within all cold-shock proteins. CspA facilitates translation at low temperature by destabilizing mRNA structures and is the major Csp in some bacteria [17,18]. In the studied biocrusts, CspA was found to be the most abundant Csp (Figure 1). Furthermore, the majority of reads annotated as CspA were identified by Kraken2 as originating from bacteria. Three specific genera, namely Hymenobacter (Bacteroidota), Polymorphobacter, and Sphingomonas (proteobacteria), were consistently detected across all the analyzed samples. The presence of CspA has been already recorded in the available genomes of these genera documented at the UniProt database. Nevertheless, 25–38% of the CspA reads could not be classified to any known taxa. This may suggest the presence of either yet-to-be-described bacterial taxa or under-represented eukaryotic organisms in the Kraken2 database. In addition, CspC and CspE are involved in regulation of expression of stress response proteins such as RpoS and UspA [4], which were also observed in the studied samples. Most of the reads annotated as CspC and E were classified under the phylum Pseudomonadota, which is consistent with previous findings of the presence of cold shock proteins in the available genome of Pseudomonas [19]. In contrast to bacteria, eukaryotic microalgae, particularly Chlamydomonas, contain only a single cold shock protein, NAB1 [5], and we did not observe it in the metagenomic datasets.

Cold shock DEAD-box protein A (CsdA) is essential for cold-tolerant and psychrophilic microorganisms as it prevent the formation of double-stranded mRNA stabilized at low temperatures and, subsequently, allows the translation of the mRNA by the ribosomal complex [20]. Between 299 and 653 CsdA were detected in the Antarctic biocrusts, and the majority originated from Actinomycetes (Figure 1; Appendix A). CsdA has been previously observed in the genomes of actinobacteria taxa [21]. 

Antifreeze proteins (AFP) are ice-binding proteins that inhibit ice crystal growth and, thus, protect the microorganism cell damage due to freezing. We reported 6–48 AFP in the collected biocrust samples (Figure 1). Actinomycetes were the only bacterial class associated with the recorded AFP (Appendix A). A few strains of Actinomycetes, specifically from the genera Subtercola and Cryobacterium, isolated from cryoconite holes in Svalbard, were previously observed to exhibit AFP activity in the culture medium [22]. However, the majority of the reads assigned as AFP were not identified by Kraken 2. These proteins were previously reported in Antarctic eukaryotic microalgae, such as Chlamydomonas and Chlorominima [6,23]. Chlamydomonas, as a typical representative of polar biocrusts [1], could potentially be the source of the unclassified AFP in this study.

Overall, the composition Csp, CsdA, and AFP in the studied biocrusts did not differ pronouncedly among the different sites. However, the biocrust sample with penguin feather (Amu14) exhibited a higher number of CsdA. Despite having a lower abundance of bacteria, Amu14 had a higher number of eukaryotic reads compared to the other samples. Considering that the amplicon sequencing showed the dominance of Chloroplastida within the eukaryotic community in this sample , it could be suggested that microalgae might be the main source of the CsdA in the Antarctic biocrusts. The presence of CsdA was confirmed in Chlamydomonas [5], but unfortunately, there has been a limited number of studies investigating CsdA genes in other eukaryotic microalgae. In addition, the similarity of the majority of the sequences to the NCBI database was below 90%, suggesting the possible occurrence of novel Csp, CsdA, and AFP in the studied biocrusts (Appendix A).

In conclusion, this study provides valuable insights into the metagenomic profile of biocrusts in Antarctica. Moreover, it lays the foundation for future investigations on the adaptation of microorganisms in extreme cold environments and highlights the importance of understanding the role of cold shock and antifreeze proteins in Antarctic biocrusts.

## Figures and Tables

**Figure 1 microorganisms-11-01932-f001:**
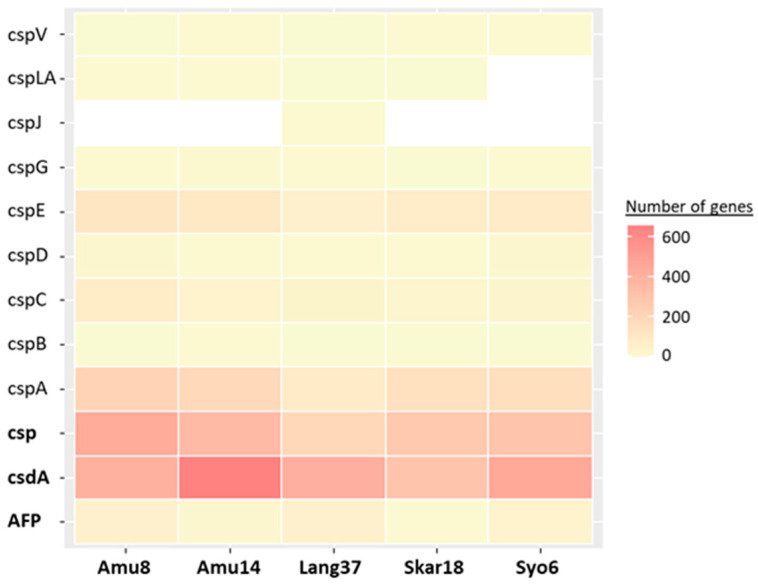
Metagenomic profile of cold-shock proteins (Csp), cold-shock domain (CsdA), and antifreeze proteins (AFP) in the biocrusts from east Antarctica. White color indicates no genes recorded in the sample.

**Table 1 microorganisms-11-01932-t001:** General overview on metagenomic sequencing of the Antarctic biocrusts.

		Amu8	Amu14	Lang37	Skar18	Syo6
**Assembly**	Number of contigs	4,077,685	3,483,827	2,978,391	2,810,569	3,662,403
	N50	1200	1228	1524	1685	1307
**GO annotation**	Blast, mapped and annotated contigs	1,206,014	1,008,266	865,031	823,442	1,107,021
	%	29.6	28.9	29.0	29.3	30.2
	EggNOG GO annotated contigs	163,852	154,537	114,524	109,335	147,282
	%	4.0	4.4	3.8	3.9	4.0
**ORFs**	Predicted ORFs	4,873,800	4,214,631	3,651,826	3,427,933	4,430,369
	Avg. gene length (nt)	369	462	384	379	373

## Data Availability

The raw reads were submitted to the Sequence Read Archive (SRA) under the project PRJNA945601.

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
