# Peer review of "Metagenomic Analysis of Antarctic Biocrusts Unveils a Rich Range of Cold-Shock Proteins"

_microorganisms, 2023, doi:10.3390/microorganisms11081932_

Round 1

Reviewer 1 Report

Summary:

This paper provides insights into the metagenomic profile of biocrusts in Antarctica. It highlights the importance of understanding the role of cold shock and antifreeze proteins in Antarctic biocrusts. The study also lays the foundation for future investigations on the adaptation of microorganisms in extreme cold environments. The paper reveals the presence and abundance of cold shock proteins (Csp), cold shock domain A proteins (CsdA), and antifreeze proteins (AFP) in these extreme environments. The majority of the Csp recorded in the studied biocrusts were Csp A, C and E. In addition, CsdA, Csp and AFP were primarily originated from Proteobacteria and Actinobacteria.

Comments:

L 50-52: Sample characteristics, sampling site etc. need to be included here. One can’t refer to unpublished works in this case.

L57: Same issue, please describe your methods in detail such as DNA extraction protocols, amount etc in detail.

L63: Please show the taxonomic assignment of each sample preferably to contemporary versions SILVA and or GTDB taxonomies.

L68-69: How did you inform cutoff thresholds and have gene annotations been confirmed using phylogenetic trees?

Supplementary:

Supplementary Figure 1: I can see three categories (Unknow, Eukaryota, Bacteria) but shown are five, where are the other two?

To lift this paper I suggest the following:

Figure 1:  Make a nice figure including a map showing sampling sites and coordinates, stacked barchart of relative abundance (based on data in Suppl Tables) and sample richness.

Figure 2: Metagenomic profile (Currently Fig.1)

Move both tables to supplementary.

Author Response

Thank you very much for your comments and suggestions!

L 50-52: Sample characteristics, sampling site etc. need to be included here. One can’t refer to unpublished works in this case.

Re: The description of the localities and sampling sites was added to the manuscript as suggested.

L57: Same issue, please describe your methods in detail such as DNA extraction protocols, amount etc in detail.

Re: The additional information was added to the Methods.

L63: Please show the taxonomic assignment of each sample preferably to contemporary versions SILVA and or GTDB taxonomies.

Unfortunately, Kraken 2 database cannot be compared to the mentioned databases, because the taxonomic assignments have different algorithm. Kraken2 examines the k-mers within a read and querying a database with those k-mers. This database contains a mapping of every k-mer in Kraken's genomic library to the lowest common ancestor in a taxonomic tree of all genomes that contain that k-mer.

L68-69: How did you inform cutoff thresholds and have gene annotations been confirmed using phylogenetic trees?

Re: We used default parameters implemented in OmicsBox software. Annotation Cut-Off is 55. Furthermore, the software employs evidence codes, which promote the assignment of annotations with experimental evidence and penalizes electronic annotations or low traceability.

Supplementary:

Supplementary Figure 1: I can see three categories (Unknow, Eukaryota, Bacteria) but shown are five, where are the other two?

Re: Supplementary Figure 1a contains five categories presented in the legend: Archaea, Bacteria, Eukaryota, Unknow, Viruses. However, the numbers for Archaea and Viruses are so low that they are not visible on the figure. You can see a hint of Archaea in the sample Amu8.

To lift this paper I suggest the following:

Figure 1:  Make a nice figure including a map showing sampling sites and coordinates, stacked barchart of relative abundance (based on data in Suppl Tables) and sample richness.

Figure 2: Metagenomic profile (Currently Fig.1)

 Re: The map of the localities is already used in our first manuscript. The revision of the manuscript has been already submitted to FEMS Microbiology Ecology. We believe that it will be accepted before this manuscript. Then we will place the citation accordingly.

Move both tables to supplementary.

Re: The Table 1 was moved to the supplementary as suggested. We would like to leave Table 2 (Table 1 in current version) in the main text.

Reviewer 2 Report

As general comment the work is well written and designed with relevant results.
The authors touch upon very important issues about the metagenomic analysis of Antarctic biocrusts .
The issues discussed by the Authors are original.
This manuscript timely and I commend the Authors for bringing in some new ideas and analysis.
The work does not raise any scientific or substantive reservations.
This study is very interesting and conforms to the requirements of the Microorganisms  journal.

Materials and method section is well described and correspond to the aim set out in the manuscript.  
The results are correctly described.
The tables and figure  are clear and understandable.
The discussion is correct.
The conclusions of the article are consistent with the problems discussed.
The references are properly chosen and cited.
The paper needs some editorial corrections.
I recommend the publication of this manuscript in the Microorganisms  journal.

Author Response

Thank you very much for a nice feedback.

Reviewer 3 Report

General comments

This communication manuscript describes the metagenomic analysis of Antarctic biocrusts and identifies the cold shock protein sequences. Although recent advances in high-throughput sequencing techniques have elucidated the microbial community structure of Antarctic ecosystems, the knowledge of their function and metabolism is limited. In this context, this study could be important, but currently there are the following tw concerns. First, much of the section on methods and background of the study relied on unpublished data from the authors (described as under revision), and the lack of detail made it difficult to evaluate this manuscript. For example, the DNA extraction method is not described, so it is unclear whether a sufficient amount of sample was used, whether replicate experiments were conducted, etc. Second, the sequences detected have been evaluated mainly for their presence or absence, but there seems to be a lack of results and discussion regarding the novelty of the sequences. The results should be complemented by information on sequence identity and/or placement on the phylogenetic tree (e.g., the sequences in this study form a different cluster than the known cold shock proteins.).

Specific comments

L107-109 and others: Scientific names should be in italics.

Table 1: How did you measure the pH and conductivity of the biocrust (soil) samples? This information should be needed for the method section.

Figure S1: What are the “Unknown” categories that dominate the metagenome and are not classified as prokaryotes, eukaryotes, or even viruses?

Author Response

Thank you very much for your comments and suggestions!

First, much of the section on methods and background of the study relied on unpublished data from the authors (described as under revision), and the lack of detail made it difficult to evaluate this manuscript. For example, the DNA extraction method is not described, so it is unclear whether a sufficient amount of sample was used, whether replicate experiments were conducted, etc.

Re: The methods were improved and more information about the sampling sites, sampling and DNA isolation was added.

Second, the sequences detected have been evaluated mainly for their presence or absence, but there seems to be a lack of results and discussion regarding the novelty of the sequences. The results should be complemented by information on sequence identity and/or placement on the phylogenetic tree (e.g., the sequences in this study form a different cluster than the known cold shock proteins.).

Re: As suggested, we created Suppl. Table 5 with BLAST top hit and similarity. The discussion of the sequence’s novelty was also added to the discussion.

The sequences retrieved from UniProt database and those from this study have different length and coverage resulting in difficulties to build the phylogenetic tree.

Specific comments

L107-109 and others: Scientific names should be in italics.

Re: The font was corrected.

Table 1: How did you measure the pH and conductivity of the biocrust (soil) samples? This information should be needed for the method section.

Re: This information was added to the text as suggested.

Figure S1: What are the “Unknown” categories that dominate the metagenome and are not classified as prokaryotes, eukaryotes, or even viruses?

Re: “Unknown” corresponds to the non-identified taxa. The databases are limited and a lot of sequence could not be taxonomically assigned.

Round 2

Reviewer 3 Report

All the concerns I raised have been addressed.